# Establishment of a Pseudovirus Platform for Neuraminidase Inhibiting Antibody Analysis

**DOI:** 10.3390/ijms24032376

**Published:** 2023-01-25

**Authors:** Yulia Desheva, Nadezhda Petkova, Igor Losev, Dmitry Guzhov, Alexey Go, Yu-Chan Chao, Chih-Hsuan Tsai

**Affiliations:** 1Virology Department, Federal State Budgetary Scientific Institution, Institute of Experimental Medicine, 197022 Saint Petersburg, Russia; 2Clinical Infectious Diseases Hospital No. 30 Named after S.P. Botkin, 195067 Saint Petersburg, Russia; 3Medical Center, St. Petersburg Research Institute of Epidemiology and Microbiology Named after Pasteur, 197101 Saint Petersburg, Russia; 4Institute of Molecular Biology, Academia Sinica, Taipei 115, Taiwan; 5Department of Entomology, National Chung-Hsing University, Taichung 402, Taiwan; 6Department of Entomology, National Taiwan University, Taipei 106, Taiwan; 7Department of Microbiology and Immunology, College of Medicine, National Cheng Kung University, Tainan 701, Taiwan

**Keywords:** influenza infection, neuraminidase-inhibiting antibodies, Sf21 cell culture, baculovirus, pseudotypes

## Abstract

Neuraminidase (NA)-based immunity to influenza can be useful for protecting against novel antigenic variants. To develop safe and effective tools to assess NA-based immunity, we generated a baculovirus-based pseudotyped virus, N1-Bac, that expresses the full-length NA of the influenza A/California/07/2009 (H1N1)pdm09 strain. We evaluated the level of NA-inhibiting (NI) antibodies in the paired blood sera of influenza patients by means of an enzyme-linked lectin assay (ELLA) using the influenza virus or N1-Bac. Additionally, we evaluated the level of NA antibodies by means of the enzyme-linked immunosorbent assay (ELISA) with an N1-expressing Sf21 culture. We detected a strong correlation between our results from using the influenza virus and NA-Bac pseudoviruses to detect NI antibodies and a medium-strong correlation between NI antibodies and NA antibodies determined by an N1-cell ELISA, indicating that baculovirus-based platforms can be successfully used to evaluate NI or NA antibodies. Furthermore, animal experiments showed that immunization with N1-Bac protected against infection with a drift variant of the A/H1N1pdm09 influenza virus. Our results demonstrate that recombinant baculovirus can be an effective influenza pseudotype to evaluate influenza serologic immunity and protect against influenza virus infection.

## 1. Introduction

Influenza virus (genus *Influenzavirus*, family *Orthomyxoviridae*) epidemics result in approximately 3–5 million severe infections and between 290,000 and 650,000 deaths worldwide annually [1], with sporadic outbreaks caused by the A/H5N1 and A/H7N9 avian influenza viruses further threatening human health and the economy. Hemagglutinin (HA) and neuraminidase (NA) are the two main surface antigens that are responsible for binding to host cell surface receptors (HA) and releasing the viral particle (NA).

The 2009 influenza pandemic caused by the A(H1N1)pdm virus sparked interest in NA subtype N1 cross-reactive antibodies. Antibodies directed against the main antigen of the influenza virus—HA—are virus-neutralizing, but do not protect against infection when viruses with a new HA subtype appear in the circulation. In this case, the presence in some people of cross-reactive anti-NA antibodies acquired through contact with previously circulating influenza viruses can be decisive in reducing morbidity and mortality [2]. In this regard, several current studies are devoted to investigating the formation and protective function of antibodies against NA during infection with influenza viruses and immunization with influenza vaccines.

Studying the antigenic properties of NA is of great scientific and practical interest. Neuraminidase-inhibiting (NI) antibodies can effectively limit transmission, reduce the severity of influenza infection, or prevent the development of secondary complications [1,2,3]. Since NA-based immunity may exert a protective effect against novel antigenic variants of the influenza virus, detecting NA antibodies is an important facet of studying levels of herd immunity against influenza viruses with a new hemagglutinin (HA) subtype, as well as for assessing the immunogenicity of influenza vaccines [4]. Seroconversion of HA antibodies is traditionally used to assess influenza vaccine immunogenicity, representing the ‘gold standard’ assay for this purpose. The World Health Organization (WHO) has repeatedly highlighted the importance of standardizing existing methods and developing new or improved ones for detecting NA antibodies [5]. The recently established NAction! Working Group brings together expert scientists and industry leaders to promote NA research to better understand how the properties of NA can aid the development of novel cross-protective vaccines [6].

In recent years, NA-inhibiting antibodies have been studied intensively due to their broad cross-reactivity and neutralizing and protective properties. However, currently, there is no standardized method for analyzing NA antibodies and no suitable platform for studying their neutralizing properties [6]. Utilizing recombinant NA that maintains its tetrameric form is one approach to studying how NA immunity is targeted and the protection provided by NA-inhibiting antibodies [7]. Another method is to express pseudotyped NA in a native functional state for serological research. In recent years, pseudotyped lentiviruses have received widespread attention for their utility in developing innovative diagnostic tools to study immunoprotective factors against influenza infection [8]. However, several drawbacks are present in handling lentivirus, the most worrisome of which is accidental exposure to laboratory workers as the vector still possesses oncogenic potential [9]. Therefore, viral vectors, e.g., baculoviruses, that are available for use at biosafety level 1 (BSL-1) facilities are attractive for the production of pseudotyped influenza.

Baculoviruses are a group of invertebrate viruses that have long been used in the production of eukaryotic recombinant proteins [10]. Baculoviruses have particular advantages over conventional mammalian vectors in that they present a large genomic capacity for foreign gene insertion, low cytotoxicity in mammalian cells even with a high multiplicity of infection (MOI), and a highly specific host range, making them some of the safest eukaryotic vectors [11]. Previously, baculoviruses had been used to develop pseudovirus for coronaviruses [12,13] and express antigenic proteins from pathogens on the surface of insect cells as a cell platform for testing antisera obtained from patients or infected animals [13,14,15].

To this end, we generated a recombinant baculovirus to express the NA of an influenza virus and demonstrated that both the recombinant baculovirus and insect cell cultures could serve as a useful platform for evaluating NA antibody contents in patients’ sera. Furthermore, the recombinant baculovirus’s potential as an influenza immunogen and vaccine platform was investigated.

## 2. Results

### 2.1. Production of a Recombinant Baculovirus Expressing Functional Influenza Virus NA

To generate a recombinant baculovirus that expresses influenza NA protein on its viral envelope, we inserted the NA sequence of A/California/07/09 (H1N1)pdm09 into a baculovirus expression vector (Figure 1A). The full-length NA protein is designed to be secreted extracellularly by a honeybee melittin (HM) signal peptide and anchored on the plasma membrane/viral envelope by the NA transmembrane domain. We designated the recombinant virus as N1-Bac. Insect cells (Sf21) infected with N1-Bac exhibited recombinant protein expression compared to wild-type baculovirus (wt-Bac)-infected cells that showed no signs of protein expression (Figure 1B). We then used these insect cells to perform a 2′-(4-Methylumbelliferyl)-α-D-N-acetylneuraminic acid (MUNANA) assay. In this assay, the MUNANA substrate is cleaved in the presence of functional NA to release the fluorescent product 4-methylumbelliferone (4-MU), which can be detected and quantified by a fluorometer. We found that insect cells infected with N1-Bac (N1-cell) exhibited comparable NA activity to N9 purified protein, as revealed by serial dilution (Figure 1C). These results indicated that recombinant N1 protein was properly expressed and bioactive in N1-Bac-infected insect cells.

We further characterize the enzymatic activity of N1 expressed on N1-Bac recombinant baculovirus. In our assay evaluating cleavage of the high-molecular-weight fetuin substrate, N1-Bac exhibited no less NA activity than a whole purified A/H7N1 influenza virus. In contrast, wt-Bac showed no NA activity (Figure 2). Since the enzymatic activity of an NA protein strongly relies on its tetrameric structure, these results indicated that N1-Bac processes recombinant N1 proteins of appropriate structure on the viral particle.

### 2.2. Development of a Baculovirus Platform for Assessing Antibodies to Influenza Virus NA

Building on the result of Figure 1 that N1 proteins are expressed on the plasma membrane of insect cells infected with N1-Bac, we developed a cell-based ELISA with these cells for NA antibody detection. This ELISA platform is specific and convenient because the cells adsorbed on plastic panels are stable during transportation and do not require a cold chain. In a preliminary test, we used human antisera collected from patients with a confirmed influenza infection or vaccinated with seasonal live influenza vaccines to test the specificity of an N1-cell ELISA. These sera have been characterized as NI-positive and NI-negative by an enzyme-linked lectin assay (ELLA). In the results, both NI-positive and NI-negative sera showed significant levels of OD_450_ in an N1-cell ELISA (Figure 3). By subtracting the absorbance of wt-cells from the absorbance of an N1-cell, NI-positive samples still showed significant ELISA values, whereas NI-negative samples almost lost all the ELISA values (Figure 3).

### 2.3. Evaluation of NA Antibodies in the Paired Sera of Patients with an Influenza Infection

To examine the feasibility of the baculovirus platform, we must have definitive N1-positive serum samples. We investigated the paired serum samples obtained from 15 patients with an RT-PCR-confirmed A/H1N1 influenza infection (Table 1).

HI and NI antibody seroconversions in the patient sera were determined using whole influenza viruses. For the HI test, we used the A/California/07/2009 (H1N1)pdm09 influenza virus. For the ELLA assessing NI antibodies, a reassortant A/H7N1 virus containing NA from the A/California/07/2009 (H1N1)pdm09 was used. As illustrated in Figure 4, a few samples showed an incremental increase in HI and NI antibodies within the first 3–4 days of hospitalization, but most of the samples had noticeably increased antibody levels after the fifth day (Figure 4C). In general, a statistically significant increase in HI and NI antibodies was noted among patients (Figure 4B,D). Seroconversion of both HI and NI antibodies occurred in 46.7% of cases, while combined conversions of HI and NI antibodies in the same pairs of sera were observed in 40% of the observed patients (Table 1). No correlation was demonstrated between HI and NI antibody titers in the analyzed patient sera: r_s_ = 0.23 (Figure 4E) or r_s_ = 0.05 in patients with ≥4-fold HI antibody seroconversions (Figure 4F).

NA-antibodies, either NI antibodies or total NA antibodies, were analyzed for the same batch of sera by the baculovirus-based platform. We first used the N1-Bac virus to perform the NI assay and obtained seroconversion results that were highly similar to those obtained with the A/H7N1 virus (Figure 5A). Next, we used an N1-cell ELISA to determine the NA antibody levels and found that most samples exhibited high NA antibody levels, especially those in which high NI antibodies had been detected (Figure 5B). In general, a statistically significant increase in antibody titers was noted for both assays (Figure 5C). We used a regression analysis to assess the correlation of NI antibody titers determined using the A/H7N1 influenza virus to NA-Bac antibodies determined by an N1-Bac-ELLA and an N1-cell ELISA. A high level of correlation (r_s_ = 0.884) was shown between NI antibody titers determined using the A/H7N1 virus and N1-bac. For NA antibody titers obtained by an N1-cell ELISA, a medium-strength correlation (r_s_ = 0.695) was shown with the NI antibody titers determined by the A/H7N1 virus (Figure 5D).

### 2.4. Protective Properties of Antibodies to Influenza Virus NA in a Mouse Model of Influenza Infection

To demonstrate how the NA antibodies acquired from the A/H1N1pdm09 influenza virus can protect against infection with a drift variant of the virus, we immunized mice with N1-Bac, wt-Bac, or PBS and determined immunogenicity in the mice. After one dose of immunization, serum IgG against the homologous influenza virus A/California/07/09 (H1N1)pdm09 and the drift variant A/South Africa/3626/13 (H1N1)pdm09 were absent in all groups. Previously, it was shown that the NA of A/California/07/09 (H1N1)pdm and A/South Africa/3626/13 (H1N1)pdm differed by 10 amino acid substitutions [16]. After the second dose of immunization, mice immunized with N1-Bac presented significant serum IgG levels for both A/California/07/09 (H1N1)pdm09 and A/South Africa/3626/13 (H1N1)pdm09 (Figure 6), indicating that N1-Bac immunization could elicit IgG against A/H1N1pdm09 viruses.

Following immunization, mice in all the groups were challenged with the A/South Africa/3626/13 (H1N1)pdm09 virus. The results showed that all the mice in the group immunized with N1-Bac survived, whereas 80% of the mice that were immunized with wt-Bac or PBS died (Figure 7A). A statistically significant reduction in weight loss was also found in the group with N1-Bac compared to the other two groups (Figure 7B). In the lungs of mice immunized with N1-bac, a 100-fold decrease in the average titers of the infecting virus was revealed compared to mice injected with wt-Bac or PBS (Figure 7C). Thus, administration of N1-Bac was shown to protect against mortality and partially protect against lung infection after challenge with a drift variant of the pandemic influenza virus.

## 3. Discussion

Antibodies to influenza virus surface antigens are one of the main factors that determine a person’s susceptibility to influenza infection. In the case of influenza virus antigenic variants with new HA subtypes, the detection of NA antibodies may be more informative than the detection of HI antibodies because antibodies to novel HA are usually absent in the population. To date, the most specific method for detecting NI antibodies is an ELLA using high molecular weight substrates—glycoproteins containing neuraminic acid—which directly measures the yield of the enzymatic reaction product in the presence or absence of a blood serum sample [17].

Generally, a reassortant virus is used in an ELLA to detect NI antibodies. It has been reported previously that monoclonal antibodies to the highly conservative HA stalk domain [18] may reduce the NA activity through steric interactions [19], thus affecting the results of the NI reaction. In order to check whether the results are NI-specific, the obtained data needs to be compared with the test results of the pure NA. However, NA in solution exists in a two-dimensional form and is inferior in enzymatic activity to tetrameric NA in its native conformation [20]. The baculovirus platform is a convenient model to demonstrate the specificity of NA inhibition by serum antibodies. A high level of correlation of NI results with the whole A/H7N1 virus and N1-Bac was shown (Figure 5D). At the same time, a medium-strong relationship was shown between the NA antibody titers obtained in the N1-cell ELISA and ELLA A/H7N1 (Figure 5D). Interestingly, a medium-strong relationship (Spearman’s r = 0.55) was also established between the data obtained using the N1-cell ELISA and the ELLA with the N1-Bac. These may be due to different fractions of antibodies detected in the ELISA and ELLA since an ELISA detects total NA IgG and an ELLA detects NI antibodies. Notably, the NA antibodies were studied for the first time in an ELISA using influenza virus NA expressed on the Sf21 cell membrane. This test is specific and quite convenient because the culture adsorbed on plastic panels is stable during transportation and does not require a cold chain. One drawback is the need for normalization with a wt-cell ELISA (Figure 3), which doubles the reagent consumption. On the other hand, the use of pseudo-typed baculovirus N1-Bac proved to be highly sensitive and specific in the ELLA with fetuin substrate, which may be able to replace the reassortant virus in relevant tests.

The recombinant baculovirus used in the study is multiplied in insect cell lines. Therefore, the difference between insect and human glycosylation is a point to be considered. Insect cells are more likely to produce terminally mannosylated proteins, whereas mammalian cells modify most proteins with biantennary and terminally sialylated glycan structures [21,22,23,24]. The different glycan structures in insect cells have been shown to affect protein therapeutic efficacies in several ways [24,25,26]. However, our animal experiments showed that immunization with N1-Bac protected the animals from the challenge of a drift variant of the pandemic influenza virus (Figure 7). The results confirmed that the enzymatic activity and antigenic structure of NA are preserved when surface-expressed on baculoviruses and demonstrated the potential of N1-Bac to be a vaccine antigen for A/H1N1pdm09 viruses.

## 4. Materials and Methods

### 4.1. Ethic Statements

In this study, we used archived blood serum samples from influenza patients with an RT-PCR-confirmed influenza A/H1N1 infection. The serum samples left over from routine studies were collected at the Clinical Infectious Diseases Hospital No 30. in Saint Petersburg, Russia, in influenza seasons 2017–2018 (15 pairs) and 2018–2019 (20 pairs). We also used the blood sera from healthy adults immunized with a commercial trivalent live influenza vaccine (LAIV) produced by ‘Microgen’ (Irkutsk, Russia) in 2018 (20 pairs). We additionally studied the sera of healthy patients 28–81 years old from the Medical Center of the Federal State Budgetary Scientific Institution “IEM” remaining from routine studies (60 serum samples). The study was approved by the Local Ethics Committee at the FSBSI “IEM” (protocol dated No. 3/19 dated 04/25/2019). Upon receiving approval from the Ethics Committee, the sera were handed over to the researchers, none of whom had access to the personal data of the patients (before analyzing the clinical data, the primary patient data were depersonalized, and the researchers could not access this). As this is a retrospective study, informed consent was not required. Nevertheless, upon admission to the hospital, all patients signed an informed consent that encompassed the use of clinic samples for scientific research.

### 4.2. Viruses

To assess serum antibodies in the enzyme-linked lectin assay (ELLA), we used the A/H7N1 reassortant virus, which inherited HA from A/horse/Prague/1/56 (H7N7) and NA from A/California/07/2009 (H1N1)pdm09 [16]. We also used A/California/07/2009 (H1N1)pdm09 and A/South Africa/3626/13 (H1N1)pdm09 influenza viruses obtained from the Virology Department.

### 4.3. Production of a Recombinant Baculovirus N1-Bac

Recombinant baculoviruses expressing influenza NA from A/California/07/09 (H1N1)pdm09 (N1-Bac) were prepared and propagated in the Sf21 insect cell line. Specifically, a synthetic DNA sequence encoding the transmembrane domain and ectodomain of A/California/07/09 (H1N1)pdm09 NA, followed by a hexameric Histidine tag, was inserted into a pAB-pEG-hhp10-HMc-6MC vector [14], from which the 6H-6MC vector element had been removed, with the NA sequence placed downstream of the honeybee melittin (HM) signal peptide. Recombinant AcMNPV (i.e., NA-Bac) was generated by co-transfecting the expression vector plasmids with a modified baculovirus genomic DNA, flashBac^TM^ ULTRA (Mirus Bio LLC, Madison, WI, USA), into Sf21 cells by TransIT^®^-Insect Transfection Reagent (Mirus Bio LLC, Madison, WI, USA). The resulting recombinant baculovirus was amplified in the Sf21 cells and isolated through end-point dilution, as described previously [14,15]. We used a wild-type baculovirus (wt-Bac) lacking recombinant protein expression, which has been well characterized in previous studies as a negative control [13,15].

### 4.4. Western Blotting Analysis

Sf21 cells were infected by recombinant viruses at an MOI of 1 and incubated for 2 days to express the recombinant proteins. The cells were collected, washed with Dulbecco’s phosphate-buffered saline (DPBS) to remove the culture medium, and lysed with RIPA Lysis and Extraction Buffer (Thermo Scientific, Waltham, MA, USA). To determine protein expression, equal amounts of cell lysates were separated by 10% sodium dodecyl sulfate-polyacrylamide gel (Omic Bio, Taiwan) and Western blotted with mouse anti-His antibody (1:2500).

### 4.5. Fluorescence-Based NA Activity Assay

Sf21 cells without virus infection, N1-Bac- and wt-Bac-infected Sf21 cells, and purified N9 protein (recombinant soluble ectodomain of H7N9 NA, purchased from Sino Biological, Beijing, China) were two-fold serially diluted (with a volume of 20 μL) in 96-well opaque black flat-bottom microplates (Corning Inc., Corning, NY, USA) using 1x NA-Flour Assay buffer (33 mM MES, 4 mM CaCl_2_). Thirty μL of 2′-(4-methylumbelliferyl)-α-D-N-acetylneuraminic acid sodium salt hydrate (4-MUNANA; Sigma-Aldrich, St. Louis, MO, USA, Cat: M8639) (final concentration of 100 μM) was added to each of the 96-well plates and mixed well by pipetting. Samples in 96-well plates were then incubated at 37 °C for 1 h to catalyze the 4-MUNANA substrate. The reaction was terminated by adding 150 μL of stop solution (0.14 M NaOH in 83% ethanol), and fluorescent signals were read using an excitation wavelength of 365 nm and an emission wavelength of 450 nm.

### 4.6. NA Activity Assay

Whole purified influenza viruses, N1-Bac or wt-Bac, were two-fold serially diluted and added to plates coated with fetuin substrate at a concentration of 50 µg/mL. The starting concentration of the influenza virus was 256 hemagglutinating units (HAU), and the starting concentration of N1-Bac or wt-Bac was 1 × 10^7^ PFU/mL. We used receptor-destroying enzyme (RDE) from *Vibrio cholerae* NA extract (Denka Seiken Co., Tokyo, Japan) at a dilution of 1:10 as a positive control. After incubating for 1 h at 37 °C, the plates were washed four times, then incubated with peroxidase-labeled peanut lectin (2.5 μg/mL, Sigma-Aldrich, St. Louis, MO, USA) for 1 h at room temperature, followed by washing and the addition of 100 μL of the peroxidase substrate 3,3′, 5,5′-tetramethylbenzidine (TMB). The reaction was stopped after 5 min by adding 100 μL of 1 N sulfuric acid. Optical density (OD) values were measured at 450 nm using a universal microplate reader (Elx800, Bio-Tek Instruments Inc., Winooski, VT, USA).

### 4.7. Enzyme-Linked Lectin Assay (ELLA)

We used an ELLA to evaluate NA-inhibiting antibodies as described previously [16]. Briefly, 96-well plates (Greiner Bio-One, Kremsmünster, Austria) were coated overnight with 150 μL of 50 μg/mL fetuin (Sigma-Aldrich, St. Louis, MO, USA). Next, 60 μL of serum was heated at 56 °C for 30 min, serially diluted with phosphate-buffered saline containing bovine serum albumin (PBS-BSA), and then incubated with an equal volume of pre-diluted virus, N1-Bac or wt-Bac, for 30 min at 37 °C. The content of the influenza virus was 128 HAU/0.1 mL, and that of the NA-Bac or wt-Bac was 3.5 × 10^6^ PFU/mL. After incubation, 100 μL of the mixtures were applied to the fetuin-coated wells. After incubation for 1 h at 37 °C, the plates were washed before assessing NA activity by incubating with peroxidase-labeled peanut lectin as described above for the NA activity assay. The titer of serum antibodies against NA was determined as the reverse dilution of the sample with 50% inhibition of NA activity. A two-fold increase in NA antibody titer after several days of hospitalization was considered significant.

### 4.8. Cell-Based ELISA

Serum samples were diluted in the antibody dilution buffer CytoVista (Invitrogen, Waltham, MA, USA) at a starting concentration of 1:4. For each serum sample, serum titration was performed simultaneously on 96-well plates seeded with Sf21 cells infected with N1-Bac or wt-Bac (i.e., N1-cells and wt-cells). Fifty μL of diluted serum was added to each well and incubated for 1.5 h at 37 °C. The plates were washed three times with PB with tween (PBT, 200 µL per well), after which 100 µL of conjugated secondary antibodies diluted to the optimal concentration (1: 4000) in an ELISA diluent (Invitrogen, Waltham, MA, USA) were added. The plates were incubated at room temperature and then washed five times with PBT. After that, 100 μL of TMB substrate solution per well was added to all wells, and after the appearance of sufficient color, 100 μL of stop reagent (1N sulfuric acid) was added to all wells. The optical density was measured on a plate photometer at a wavelength of 450 nm (OD450). In data analysis, the absorbance obtained with wt-cells was subtracted from the absorbance of N1-cells for each serum sample to obtain the final ELISA titer.

### 4.9. Hemagglutination Inhibition (HI) Test

Serum HI was performed using a 0.75% human erythrocyte suspension (“0” group) in 96-well U-bottom polymer plates, as described previously [27]. The sera we studied were treated with RDE and heated at 56 °C for 30 min. Each serum (tested in duplicate) was serially diluted from an initial serum dilution of 1:5 or 1:10 and mixed with eight agglutinating units (AU) of the A/California/07/2009 (H1N1)pdm09 influenza virus. Antibody titers were expressed as the reciprocal of the highest serum dilution at which inhibition of agglutination was observed. A 4-fold or more increase in antibody titer was considered a significant conversion of HI antibodies [28].

### 4.10. Mouse Immunization and Challenge

Groups of female BALB/c mice (n = 25) received intramuscularly (IM) (1) pseudoviral particles based on baculoviruses with a surface-exposed influenza virus NA, NA-Bac, or wt-Bac at a concentration of 1 × 10^6^ PFU/100 µL, mixed with incomplete Freud’s adjuvant in a ratio of 2:1; (2) unmodified baculoviruses (Bac-wt) with incomplete Freud’s adjuvant; or (3) a mixture of sterile phosphate buffer (PB) and incomplete Freud’s adjuvant. Immunization was carried out twice, with an interval of two weeks. Two weeks after primary immunization and revaccination, blood samples were collected from the submandibular vein of each group. Serum IgG levels were detected by an ELISA.

A challenge was performed with the A/South Africa/3626/13 (H1N1)pdm09 virus at a dose of five 50% mouse lethal doses (MLD_50_). The introduction was carried out intranasally, under ether anesthesia, in the form of a solution in phosphate buffer. Each mouse received 50 μL of the solution, which was evenly distributed over both nasal passages. After infection, the weight and lethality of the animals were monitored for 14 days.

### 4.11. Statistical Analysis

Statistical processing of the results was carried out using the GraphPad software (San Diego, CA, USA). Antibody levels were reported as GMT values. For statistical analysis, antibody titers were expressed as log2 of inverse final dilutions. A Wilcoxon matched-pairs test was used to compare two dependent variables. A nonparametric measure of the statistical relationship between two variables was performed using Spearman’s rank correlation coefficient (r). A *p*-value < 0.05 was considered statistically significant.

## 5. Conclusions

Our results demonstrate that the NA protein of an influenza virus can be properly expressed on the surface of baculoviruses. These recombinant baculoviruses can serve as pseudoviruses to reliably assess NI antibody titers in an ELLA, produce a cell-based ELISA system for NA antibodies, and become easily produced vaccine antigens.

## Figures and Tables

**Figure 1 ijms-24-02376-f001:**
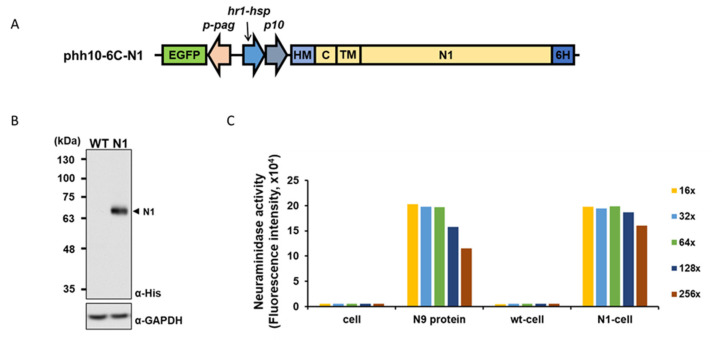
Construction and characterization of recombinant baculovirus N1-Bac. (**A**) Baculovirus expression construct for N1-Bac (phh10-6C-N1). EGFP, green fluorescent reporter; *p-pag*, *pag* promoter; *hr1-hsp70* and *p10*, a dual promoter for N1 expression; HM, honeybee melittin signal peptide; C, cytoplasmic tailed domain; TM, transmembrane domain; N1, N1 ectodomain; 6H, a hexameric histidine tag. (**B**) Western blot analysis of insect cells infected with wt-Bac and cells infected by N1-Bac. N1 was detected with an anti-His antibody. (**C**) NA activities of insect cells only (cell), purified N9 protein, insect cells infected with wt-Bac (wt-cell), and cells infected with N1-Bac (N1-cell) were detected using a fluorescence-based NA activity assay. Colors represent serial two-fold dilution factors.

**Figure 2 ijms-24-02376-f002:**
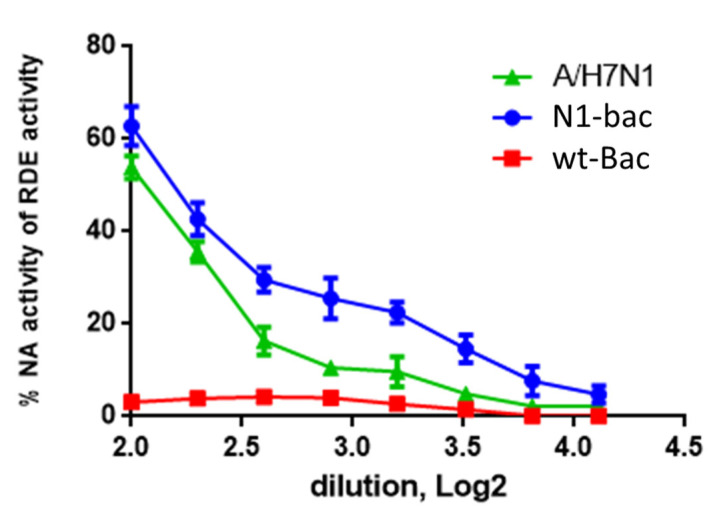
NA activities of whole purified A/H7N1 virus, N1-Bac, and wt-Bac. All samples were two-fold serially diluted and added to plates coated with 50 µg/mL of fetuin substrate. After incubation at 37 °C for 1 h, the fluorescing products generated upon NA digestion were measured using an excitation wavelength of 365 nm and an emission wavelength of 450 nm.

**Figure 3 ijms-24-02376-f003:**
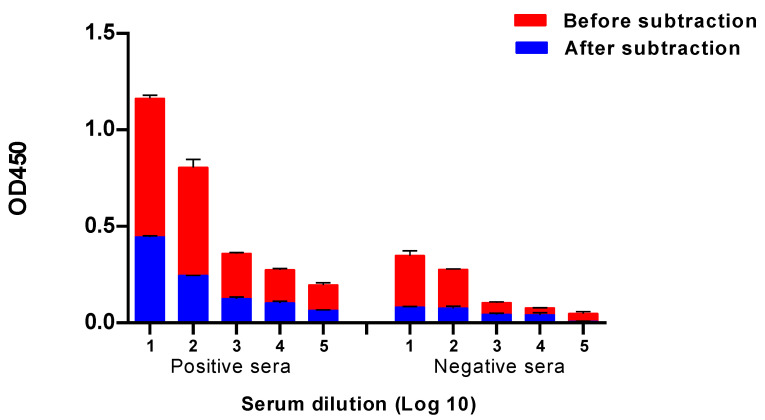
ELISA results with an N1-cell ELISA and the background subtraction of a wt-Bac-infected cell ELISA. The data from one of three independent experiments is provided.

**Figure 4 ijms-24-02376-f004:**
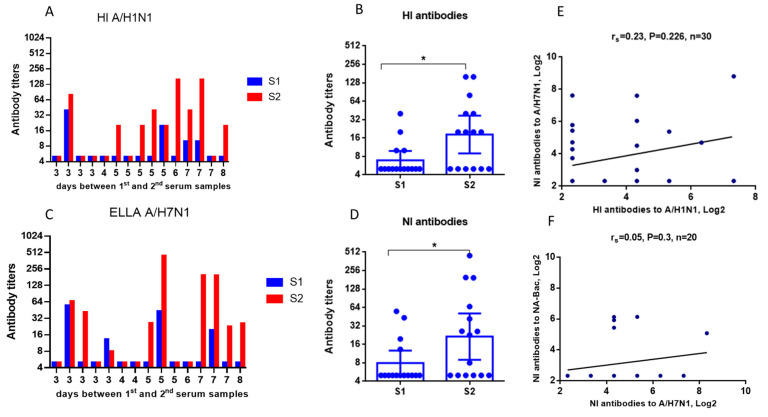
Results of the assessment of HI and NI antibodies in the paired sera of patients with a confirmed influenza infection (n = 15). S1—sera obtained on admission to the hospital. S2—sera obtained on days 3–8 after hospitalization. (**A**) HI antibody titers to A/California/07/2009 (H1N1)pdm09 influenza virus in individual sera. (**B**) Average HI antibody titers. Each dot represents an individual serum. Data are represented as geometric means with a 95% CI. *—*p* < 0.05. (**C**) NI antibody titers in individual sera determined by ELLA using the A/H7N1 influenza virus. (**D**) Average NI antibody titers. Each dot represents an individual serum. Data are represented as geometric means with a 95% CI. *—*p* < 0.05. (**E**) Correlation analysis of HI and NI antibody titers (Spearman’s r). (**F**) Correlation analysis of HI and NI antibody titers (Spearman’s r) among patients with ≥4-fold HI antibody seroconversions.

**Figure 5 ijms-24-02376-f005:**
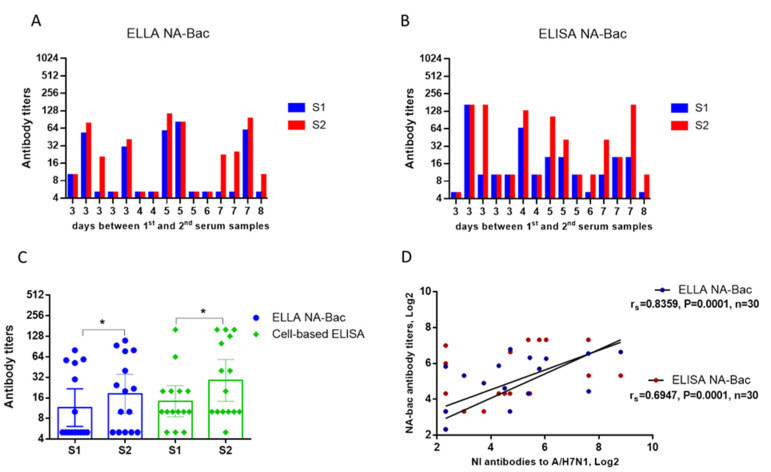
NA antibody titers of the paired sera from patients (n = 15) with a confirmed influenza infection were determined by an ELLA using N1-Bac and an N1-cell ELISA. S1—sera obtained on admission to the hospital. S2—sera obtained on days 3–8 after hospitalization. (**A**) NI antibody titers in the individual sera determined by an ELLA using N1-Bac. (**B**) NA antibody titers in the same batch of sera were determined by an N1-cell ELISA. (**C**) Mean geometric titers of NI and ELISA antibodies. Each dot represents a subject (n = 15). Dashes indicate GMT values. *—*p* < 0.05. (**D**) Correlation analysis of NI antibody titers determined using A/H7N1 influenza virus to NA-Bac antibodies determined by N1-Bac-ELLA and N1-cell ELISA (Spearman’s r).

**Figure 6 ijms-24-02376-f006:**
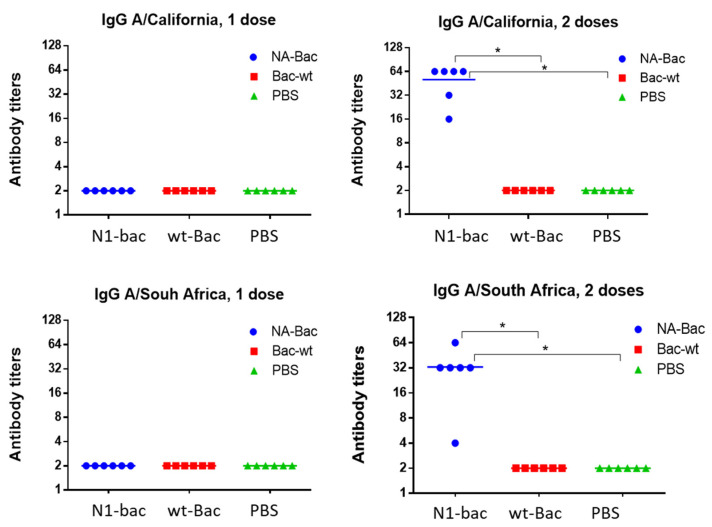
Serum IgG levels to the homologous influenza virus A/California/07/09 (H1N1)pdm09 and the drifted variant A/South Africa/3626/13 (H1N1)pdm09 (n = 6). The results of an ELISA with blood sera were obtained 2 weeks after the 1st and 2nd intramuscular immunizations. *—*p* < 0.05.

**Figure 7 ijms-24-02376-f007:**
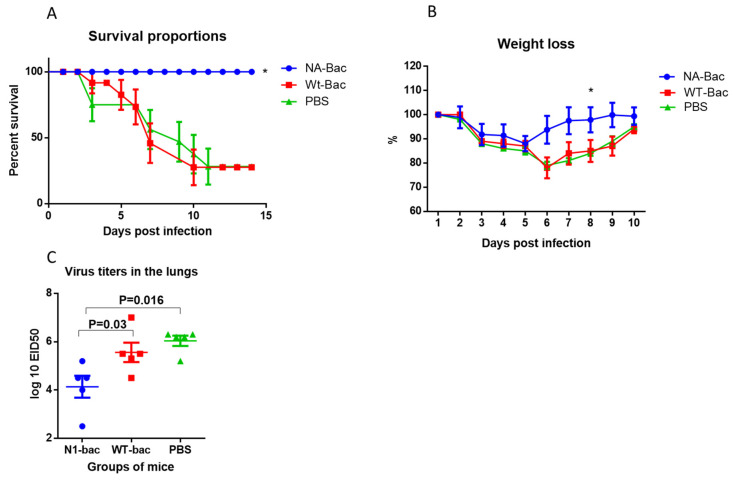
Protection after immunization with NA-bac against infection with a drift variant of the A/H1N1 pdm09 influenza virus, A/South Africa/3626/13 (H1N1)pdm09. Data from one of two independent experiments is presented. (**A**) Mortality analysis (n = 10). Log-rank (Mantel-Cox) test, ***—***p* < 0.05. (**B**) Dynamics of weight, % (n = 10). ***—***p* < 0.05. (**C**) Infectious virus isolation from the lungs (n = 5). Each dot represents an individual lung titer (EID_50_/0.1 mL). The dashes represent the mean and standard error.

**Table 1 ijms-24-02376-t001:** Patient’s characteristics.

Group Score	Influenza Infected Patients
Number in group	15
Timing of sampling	13 January 2018–9 February 2018
Mean time between first and second blood draw (days)	4.8
Age (Me; Q1, Q3)	43.00 (22.00; 50.00)
Male	8 (53.3)
Female	7 (46.7%)
4-fold HI seroconversions to H1	7 (46.7%)
2-fold NI seroconversions N1	7 (46.7%)
Both HI and NI seroconversions	6 (40%)

## Data Availability

All data is contained in the text of the paper and Appendix A file.

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
