# Peer review of "Establishment of a Pseudovirus Platform for Neuraminidase Inhibiting Antibody Analysis"

_ijms, 2023, doi:10.3390/ijms24032376_

Round 1

Reviewer 1 Report

 Summary: This manuscript describes a study to evaluate a Baculovirus-expressed influenza neuraminidase subtype 1 as a psuedovirus system to facilitate detection of antibodies against N1 and N1 inhibitory antibodies in patient sera. Correlation was observed between the baculovirus system and influenza virus for detecting inhibitory antibodies and also insect cell expressed NA to detect antibodies by ELISA. Immunization of mice with two doses the NA expressing baculovirus resulted in seroconversion to NA, presence of inhibitory antibodies and clinical and survival benefit against a strain of influenza described to possess antigenic drift.

General comments: The manuscript is of interest for its approach to develop a new reagent/system with applications for diagnostic testing, qualitative and quantitative evaluation of antibodies to N1 and as a potential to generate immunogens/vaccines. The manuscript is clear and easy to follow, and overall gives a good appraisal of the findings and limitations. I would like to see the authors provide more detail/information to explain why cross-reactivity/cross-protection is challenging for influenza viruses, addressing antigenic shift and drift to set up for the reader the significance of why they look at particular aspects.

Specific (line) comments:

26: immunization “with” N1 Bac

44-46: reference/evidence for this?

88: “as an influenza immunogen and vaccine platform”

90-92: delete text

100: remove ‘i.e.’

Figure 1: It would be good to state that phh10-6C-N1 is the construct name, and in legend state that N1 is being detected with anti-His antibody

Figure 2: what does RDE stand for?

Figure 3: “subtraction” not stbstraction. The high background from baculovirus itself is addressed in the discussion, but is the reactivity in negative sera dose dependent with dilution, and what does this tell you about the assay? Can you think of other ways this could be addressed such that you can have a better application of the system?

161-162: unclear what this means

201: drift variant – how different at nucleotide/amino acid level, or how distinct serologically. How is this characterized as a drift variant? Need to given additional information here.

202: “immunogenicity in mice”

214: “following immunization”

207: this doesn’t provide evidence that it protects against pan H1N1 viruses, just this one. Need to adjust conclusion here, or provide more data that demonstrates the breadth against antigenically diverse viruses

Author Response

We thank the Reviewer for such a detailed analysis of our paper.

Point 1. 26: immunization “with” N1 Bac

Response 1: We thank the reviewer for this correction. Corrected.

Point 2. 44-46: reference/evidence for this?

Response 2: We thank the reviewer for this remark. We have added the corresponding literary reference.

Point 3. 88: “as an influenza immunogen and vaccine platform

Response 3: We are very grateful to the reviewer for this correction. We have corrected the text.

Point 4. 90-92: delete text

Response 4: We thank the reviewer for this correction. Unnecessary text has been removed.

Point 5. 100: remove ‘i.e.’

Response 5: We thank the reviewer for this correction. Removed.

Point 6. Figure 1: It would be good to state that phh10-6C-N1 is the construct name, and in legend state that N1 is being detected with anti-His antibody

Response 6: We express our gratitude to the reviewer for this addition. We have corrected the Figure legend.

Point 7. Figure 2: what does RDE stand for?

Response 7: We thank the reviewer for this remark. In the “Materials and methods” section, we added: “We used receptor-destroying enzyme (RDE) from Vibrio cholerae NA extract (Denka Seiken Co., Japan) at a dilution of 1:10 as a positive control”.

Point 8. Figure 3: “subtraction” not stbstraction. The high background from baculovirus itself is addressed in the discussion, but is the reactivity in negative sera dose dependent with dilution, and what does this tell you about the assay? Can you think of other ways this could be addressed such that you can have a better application of the system?

Response 8: We thank the reviewer for this remark. We have corrected the designations in the figure. The fact that “seronegative” sera showed some background values, although much lower than seropositive ones, is explained by the fact that these were human sera. People other than young children are rarely completely seronegative. We determined the seropositivity of sera by inhibition of the enzymatic reaction with one specific virus. It is possible that these sera also had a low level of cross-reactive antibodies, but we did not study this, since this was not the subject of the study.

Point 9. 161-162: unclear what this means

Response 9: We thank the reviewer for this remark. We added: “in the same pairs of sera”

Point 10. 201: drift variant – how different at nucleotide/amino acid level, or how distinct serologically. How is this characterized as a drift variant? Need to given additional information here.

Response 10: We thank the reviewer for this remark. We added:Previously, it was shown that the NA of A/California/07/09 (H1N1)pdm and A/South Africa/3626/13 (H1N1)pdm differed by 10 amino acid substitutions [16]”.

Point 11. 202: “immunogenicity in mice”

Response 11: We thank the reviewer for this correction. Corrected.

Point 12. 214: “following immunization”

Response 12: We thank the reviewer for this correction. Corrected.

Point 12. 202: “immunogenicity in mice”

Response 12: We thank the reviewer for this correction. Corrected.

Point 13.  207: this doesn’t provide evidence that it protects against pan H1N1 viruses, just this one. Need to adjust conclusion here, or provide more data that demonstrates the breadth against antigenically diverse viruses

Response 13: We thank the reviewer for this correction. It was a typo and has now been removed. “…indicating that N1-Bac immunization could elicit IgG against A/H1N1pdm09 viruses”.

Reviewer 2 Report

the authors designed a construct to express influenza N1 on baculovirus.  Western blot and NA activity assays were used to confirm the surface expression. Based on the baculovirus platform they can also do ELISA to assess NA antibodies. They demonstrated that this system can be used to test seroconversion of influenza infected patients. Finally they showed the baculovirus platform can be used as a vaccine. 

Overall the paper is an interesting paper for immunology field and I support publish the paper. But I hope the authors could address the following issues: 

1. line 90-92 should be removed 

2. the authors should also do a Correlation analysis of HI and NI antibody titers but exclude those patients with out seroconversion, and add it as Figure 4F. 

3. the author should give the unit for x axis in Figure 7C, i think it should be days

Author Response

The authors thank the reviewer for his kind attitude towards our work and for all the corrections.

Point 1. line 90-92 should be removed 

Response 1: We thank the reviewer for this correction. Corrected.

Point 2. the authors should also do a Correlation analysis of HI and NI antibody titers but exclude those patients without seroconversion, and add it as Figure 4F. 

Response 2: We thank the reviewer for this remark. We have added the results of the analysis in Fig. 4F

Point 3. the author should give the unit for x axis in Figure 7C, i think it should be days

Response 3: We thank the reviewer for this correction. Corrected.

Reviewer 3 Report

The authors have generated a baculovirus-based pseudotyped virus expressing the full-length influenza N1 neuraminidase and performed an interesting complex research regarding the NA-based immuninty. The results are sound, well presented and contribute to the NA evaluation as a protective antigen and its activity standartization during influenza vaccine manufacturing.

There is only one methodological issue that needs to be clarified. Does the "N9 protein" used in some experiments mean the N9 subtype influenza NA? Please describe it in more detail: is it a recombinant soluble ectodomain or a full-length protein containing transmembrane domain and cytoplasmic tail? A reference or a full protocol of the N9 NA preparation should be provided.

Also, please remove the text "This section may be divided by subheadings. It should provide a concise and precise description of the experimental results, their interpretation, as well as the experimental conclusions that can be drawn." from the beginning of the Results.

Author Response

The authors thank the reviewer for his kind attitude towards our work and for all the corrections.

Point 1. There is only one methodological issue that needs to be clarified. Does the "N9 protein" used in some experiments mean the N9 subtype influenza NA? Please describe it in more detail: is it a recombinant soluble ectodomain or a full-length protein containing transmembrane domain and cytoplasmic tail? A reference or a full protocol of the N9 NA preparation should be provided.

Response 1: We thank the reviewer for this remark. The N9 protein referred to is the recombinant soluble ectodomain of H7N9 NA purchased from Sino Biological, Beijing, China. We have added this information to the Materials and methods section (4.5).

Point 2. Also, please remove the text "This section may be divided by subheadings. It should provide a concise and precise description of the experimental results, their interpretation, as well as the experimental conclusions that can be drawn." from the beginning of the Results.